# Fire, Rain and $CO_2$: Potential Drivers of Tropical Savanna Vegetation Change, with Implications for Carbon Crediting

Greg Barber [1,*], Andrew Edwards [2] and Kerstin Zander [1]

1   Northern Institute, Charles Darwin University, Darwin, NT 0909, Australia
2   Darwin Centre for Bushfire Research, Research Institute for the Environment and Livelihoods, Charles Darwin University, Darwin, NT 0909, Australia
*   Correspondence: gregory.barber@students.cdu.edu.au

**Abstract:** A global trend of increasing tree cover in savannas has been observed and ascribed to a range of possible causes, including $CO_2$ levels, changing rainfall and fire frequency. We tested these explanations in the Australian tropical savanna, taking 96 savanna 'cool burning' projects from Australia's emissions offset scheme as case studies. We obtained readings of tree cover and explanatory variables from published remote sensing or spatial data sources. These were analysed using time-series linear regression to obtain coefficients for the influence of severe fire occurrence, annual rainfall and prior percentage tree cover. Although statistically significant coefficients for the key variables were found in only half (severe fire) or one quarter (rainfall) of the individual project models, when comparing all the model coefficients across the rainfall gradient, ecologically coherent explanations emerge. No residual trend was observed, suggesting rising $CO_2$ levels have not influenced tree cover over the study period. Our approach models tree cover change by separating ecological drivers from human-controlled factors such as fire management. This is an essential design feature of national emissions inventories and emissions offsets programs, where crediting must be additional to the expected baseline, and arise from human activity.

**Keywords:** tropical savanna; woody thickening; $CO_2$ fertilisation; fire severity; offsets; emissions credits





## 1. Introduction

A global trend in 'woody thickening', or increasing tree cover in savannas and woodlands has been observed [1] and ascribed to a range of causes, from rising $CO_2$ levels [2–4], to changing rainfall and fire frequency [4–13] and the impact of large herbivores [14].

In Australian tropical savannas, in the absence of traditional aboriginal burning practices, uncontrolled fire of higher severity occurs late in the dry season [15]. After the reinstatement of the practice of early dry season burning, fire intensity and potential fire frequency have been shown to be reduced, and may lead to an increase in the amount of carbon stored in living woody biomass [16]. However, the difficulty of separating the effects of human intervention from the variation in the natural baseline, or other possible trend drivers, makes it challenging to measure this additional carbon for emissions inventory or project crediting purposes [17–21].

A range of field studies have examined the dynamics of tropical savanna vegetation in Australia. Murphy et al. [11] analysed tropical savanna vegetation change via three on-ground surveys at five-yearly intervals in three Northern Territory national parks. Direct measurements of live tree basal area were converted via an allometric to tree carbon mass. Fire intensity and frequency were recorded and, along with gridded annual rainfall and site landform, were used as explanatory variables. The study observed very little change in total biomass at the sites over time. However, fire frequency and severity were found to be influential on the variation across the sites, causing a reduction in biomass predominantly in existing trees, rather than via impacts on mortality or recruitment. Frequent and severe

fire was also observed to reduce tree growth increment in an earlier ten year study of plot-level measurements in Australian tropical savannas [22].

Using a LIDAR measure of savanna tree height and cover, lower above-ground biomass was observed in plots with higher fire intensity histories, [23]. Woody cover levels in 4000 ha of savanna in Kakadu National Park were studied [24] using aerial photographs between 1950 and 2016, and matched to a long-term record of fire history. No overall trend in woody cover over time was found at the site. Short term variation in cover was positively related to rainfall in the previous 12 months, and there were weak negative effects of fire occurring in the year of sampling and the previous 4 years. Tree cover expansion in tropical savannas in Litchfield National Park was observed over a 50-year period, from the interpretation of aerial photographs [4]. However, the study was unable to ascribe specific ecological factors to this observation. Tree cover changes over 40 years were found to track annual rainfall patterns [8] and, after controlling for rainfall, reduced tree cover was associated with higher fire frequency. In another approach, direct measurement of biomass by field sampling was compared with flux-tower measurements of gas exchange at a small number of sites in Australian tropical savannas [25] to estimate the drivers of carbon accumulation through net biome productivity

Rising atmospheric $CO_2$ concentrations have been proposed as a driver of savanna tree cover expansion [2,3,26]. One possible mechanism is that steadily increasing $CO_2$ levels are enhancing the water use efficiency of woody plants [27]. In that circumstance, the relationship between woody cover levels and annual rainfall would be expected to diverge over time, that is, tree cover might rise relative to the precipitation quantity. Contrasting this effect, it has been proposed that higher woody cover creates increasing competition between trees for available moisture [12,28], which may lead to the levels of tree cover increase reaching an upper limit.

Competition between trees and grasses are sometimes discussed as drivers of alternative stable states in savannas [29–32], with the implication that cover of either life form, once established, is likely to persist. A review of this extensively studied question [33] proposed a model of fire frequency and average rainfall 'parameter space' which could create stable states, or 'bi-stability', dependent on initial conditions and disturbance history. Sankaran et al. [34] makes the important observation that, in African savannas, rainfall may be the controlling variable for woody cover in arid and semi-arid zones, whereas periodic disturbances such as fire and herbivory become more important in 'mesic' savannas above 650 mm annual rainfall.

Factors generally not considered in this group of studies are the fire history of the sites, and the variation in topography or soil condition. All of these may impact on the potential for additional carbon storage after fire regime change [35,36].

Emissions crediting methodologies based on fire management in savannas have been implemented since 2015 under Australia's emissions 'offset' scheme [37]. The program now covers around half of the area of the tropical savanna ecosystem [15] and has produced 12.9 million tonnes $CO_2$-e of emissions abatement to date [38]. However, these methods only credit avoided emissions, where reduced fire intensity produces less of the more potent greenhouse gases, such as CH4 and $N_2O$, and more recently, the accumulation of carbon in coarse ground fuels. It has been estimated [39] that the unmeasured annual sequestration benefit from Australia's program could be 3.5 times the amount currently being credited. A recent study [40] used a demographic model derived from observed changes in field plots to predict increases in tree biomass, with an estimated crediting value three times that produced under the current method. Land-based crediting methodologies often use a baseline and credit approach [41]. The baseline is not the quantity of carbon present at the project site, but rather the 'counter-factual' that would have occurred over the project period, in the absence of human intervention.

The aim of this study is to derive an explanatory model that separates the possible drivers of changing woody vegetation cover in Australian tropical savannas [42] and generates ecologically coherent coefficients for these effects. Using insights from the

literature, we collect observations or proxy variables for the important ecological factors, and construct time-series models for each of the registered savanna burning projects using stepwise regression. The objective is to isolate the effects of the fire management program from the 'natural' or baseline drivers. Where an increased living biomass is observed and can be demonstrated to be the result of a program of planned fire management [43], separate from the influence of climatic and climate-change variables, it may be possible to credit the additional carbon.

## 2. Materials and Methods

### 2.1. Study Sites

Our test sites were the 96 approved offset projects in the Australian tropical savanna ecoregion (Figure 1). The properties are under a range of tenures; national parks, indigenous management, private conservation reserves and pastoral leases. A total of 17 have had their project status revoked and 15 of the active projects had not been issued any offset credits during the study period.

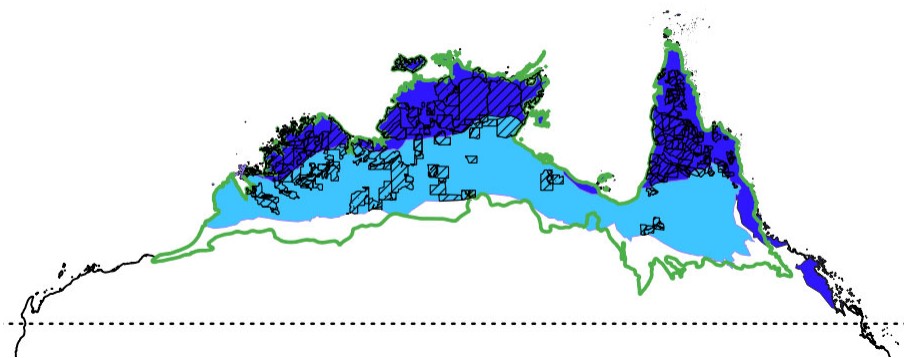

**Figure 1.** Map of the study area in tropical northern Australia, indicating Tropic of Cancer (dashed line), boundary of tropical savanna ecoregions from Terrestrial Ecoregions of the World [42] (green line), 600–1000 mm average annual rainfall zone (light blue), greater than 1000 mm (darker blue), registered savanna burning project sites (hatched areas).

The tropical savanna ecoregion [42] encompasses 1.2 million km$^2$. Average annual rainfall is observed to increase when moving northwards, as woody vegetation cover increases [44]. Fire frequency, in broad terms, increases along the same gradient, varying from decadal to almost annual in the highest rainfall zones [45].

### 2.2. Data Collection

Polygons were obtained for each of the 96 offset project property boundaries [46]. Emissions crediting may occur for burning activities on a subset of land known as the Carbon Estimation Area, however, this boundary is not available to us. These polygons were used within QGIS [47] to sample the model explanatory variables which are provided in the form of raster layers, as detailed in Table 1 below.

The dependent variable in this study was the percentage or 'fractional cover' of photosynthetic vegetation [48] derived from Landsat data, now freely available in a series since 1987 at the 25 × 25 m pixel scale. As an estimate of woody vegetation, this study used the quarterly tile from September to November. In this region, grassy and herbaceous vegetation has dried off and is largely non-photosynthetic during this period, prior to the arrival of wet season rains. Changes in green woody vegetation cover levels may indicate either expansion or contraction of canopy cover by established trees, or recruitment or loss of new woody seedlings.

**Table 1.** Description, notes and referenced sources of data used in model variables.

| Variable | Description, Notes and References |
|---|---|
| Green woody cover level during each time period | Fractional green vegetation cover data (mean % across project site) 1988–2021 [48]. Provided as seasonal (i.e., quarterly) rasters with bands for percentage cover of photosynthetic and non-photosynthetic vegetation, and bare ground. We selected the September–November quarter which is the late dry season, when most herbaceous vegetation is non-photosynthetic and we infer that the greenness is predominantly photosynthetic woody vegetation. These quarterly tiles approximate median values for the quarter, prepared using the medoid method [49]. |
| Severe fire % of area burnt for the calendar year time period | Fire severity annual area (% of project site) 2003–2019. This is available as rasters prepared as part of a previous study. Severe fire is defined as greater than 50% canopy scorched [50]. |
| Green woody cover level during each previously corresponding time period | The green woody cover measure is expected to be correlated with its level from earlier periods. We used a one-year lagged time period. |
| Rainfall for the time period | Monthly rainfall obtained from the Bureau of Meteorology [51] annualised to the time period from September of the previous year to August of the following. Raster Calculator in QGIS was used to prepare these new variables. |
| Yearly counter | A variable increasing by one integer for each annual time period. Used to estimate any remaining linear trends after regressing for the above variables. In particular, this could represent the effects of rising atmospheric $CO_2$ concentrations as over the study period $CO_2$ ppm is a mostly linear trend. |

Data on fire severity [52] is available in spatial files for Australia's tropical savannas for the period from 2003 to 2018, which limited our time-series model to 16 data points. This data set provides a classification of each pixel as severely or non-severely burnt, or unburnt, based on the extent of scorching of the tree canopy, which may influence photosynthetic potential and even tree mortality [52]. We preferred this to a severity estimate based on time of year [53] as it relies on direct observation of fire effects, and our testing of the data showed that severe fires, as measured by canopy scorch, can occur in all parts of the dry season [54]. Although this is a somewhat arbitrary cutoff and binary classification, there are technical difficulties with segmenting less severe fires into subclasses [54]. Remote sensing is from MODIS where pixels are approximately 100 times the size of the Landsat fractional green cover data, however, we expect these differences to have little effect considering the large area of the project sites.

An annual and long-term average rainfall spatial data set is provided by the Bureau of Meteorology [55]. These 'gridded' products extrapolate between weather recording stations to allow an estimate of rainfall at any point of the landscape. In the tropical savanna region, there are relatively few such stations compared to densely populated areas. This introduces a potential source of residuals when using gridded data to assess a more localised study site.

### 2.3. Statistical Analysis

The dependent variable observed annual change in tree cover in a linear regression time-series [56] model, regressed against potential drivers of tree cover change, such as variation in the extent of severe fire and annual rainfall. An annual time counter was used as a proxy variable for rising $CO_2$ levels or other time trends. Linear regression with the *lm* function and model diagnostics (packages acf, trend, Kendall [57]) were carried out in R Studio v 2022.12.0 [58]. The structure of these models is described in Table 2 below.

**Table 2.** Model structures illustrating stepwise regression approach.

|  | Outcome Variable Green Woody Cover Versus: | Model Purpose |
|---|---|---|
| Model 1 | Time trend counter | To assess if a linear trend in tree cover is observable over the period 2001–2021 |
| Model 2 | Severe fire percentage | To assess if severe fire correlates with changes in tree cover (2003–2018) |
| Model 3 | Severe fire percentage and previous year's green woody cover (i.e., lagged dependent variable) | As above, but testing for the effects of autocorrelation or persistence in tree cover |
| Model 4 | Severe fire percentage and previous 12 months rainfall | Adding the effects of recent rainfall to fire effects, to observe any interaction between their coefficients or change in model fit |
| Model 5 | Severe fire percentage and previous year's green woody cover and previous 12 months rainfall | Model with all three variables' effects |
| Diagnostic | Autocorrelation and linear trend tested against residuals from models | |

We also tested for a correlation with previous percentage green woody cover. Woody vegetation is somewhat persistent, with a life history based on investment of energy in growth and maintenance of woody tissue. A low positive coefficient would indicate that after any change or disturbance, the level of cover is likely to return to its previous level. A high positive coefficient approaching a value of 1 would indicate a strong tendency of woody vegetation cover levels to reflect last year's level, possibly indicating a 'stable state' mediated by disturbance, as described in the literature review. In statistical terms, this latter behaviour is known as a 'random walk'. Such time-series concepts have only recently been used to model forest growth [59,60]. We diagnosed the benefit of this approach by testing the residuals of the model for autocorrelation [56] using the acf function in R Studio, to test whether the current level of green woody cover shows a correlation with its own past levels at a range of time lags. Failure to account for this effect could lead to an overstatement of the statistical significance of the models, and an incorrect estimate of the models' other coefficients.

This explanatory approach to model structure is designed to accurately separate the human and environmental influences, but contrasts with a predictive model [61], which aims for more precision. Having selected the group of model variables that produces the best overall results for the whole group of projects, for consistency, these were included in all project models. Retaining terms with low statistical significance necessarily reduces the proportion of variation explained as measured in adjusted-R squared ($R^2$) terms. We tested for the Variance Inflation Factor during this process to avoid risks of multicollinearity between variables.

Climate change may be causing a trend in rainfall conditions in the study area to become more, or less, favourable to woody vegetation growth. This is controlled when annual rainfall is added to the model. However, if increased tree cover creates more competition for available soil moisture, then the effects of rainfall over time would decrease at sites where tree cover increased. This was tested for in the time-series with a logarithmic transformation added to the rainfall variable. While it has been proposed that rainfall is increasing in Australia's tropical savanna region [62–64], detecting such a trend from meteorological data would be challenging over our 20-year analysis period and is beyond the scope of this study.

It is also possible that rising $CO_2$ levels are increasing the water use efficiency of plants, in which case, tree cover and rainfall levels could show diverging trends and lower

significance in the coefficients derived. This study, therefore, tests the residuals of these models for linear trends, to capture the possibility of rising water use efficiency or other effects of rising atmospheric $CO_2$. We used a simple annual counter, not actual $CO_2$ levels, as these are known to be influenced on a global scale by the strength of the savanna sink [65–67]. Australian savannas were strongly influential on this variation [68] in line with the El Nino Southern Oscillation, which would potentially reverse the causality of $CO_2$ levels with tree cover. Because this linear counter will show some degree of correlation with any other trending variable, we chose not to include it in the model where it would affect other variables, but rather to test it against the residuals of the derived project models. This linear trend approach may be sensitive to outliers in the starting and ending values of the time-series. Therefore, we also applied the Kendall and Sen's slope tests to these residuals.

The R-squared measure almost always increases when extra variables are included in a model, so the adjusted R squared is used to correct for the inclusion of non-significant variables.

An alternative approach would be a mixed effects model combining all projects' data with crossed grouping/random effects for projects, to explain part of the overall variation in tree cover, and years, to explain any region-wide annual variation in rainfall or fire occurrence. However, regulators and offset credit buyers are likely to require that individual projects demonstrate measurable, additional carbon sequestration, so we prefer to model each project as an individual time-series and present summarised results.

## 3. Results

### 3.1. Demonstration Time-Series Data

Figure 2 below illustrates the key variables tested in our models, using the long-running West Arnhem Land Fire Abatement project area as an example. Visual inspection suggests that annual rainfall and green woody cover show related annual movements, however, later in the time-series, green woody cover appears to continue rising even as rainfall decreases. A planned, the early dry season burning program led to a reduction in the extent of severe fire [69], which can be seen in the years after the project commenced in 2005. The year 2010 saw the breaking of a long-term ENSO-driven drought and the wet season began early, impacting on green woody cover in the late dry season of the previous year, hence the apparent increase in greenness in 2009. These timing effects can create residuals which reduce the significance of the variable's coefficient.

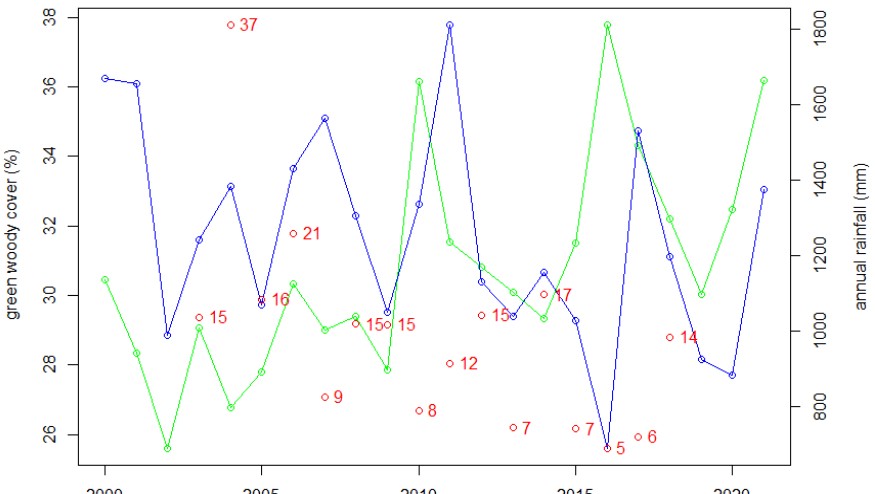

**Figure 2.** West Arnhem Land Fire Abatement (WALFA) project time series. Green woody cover, in percentage, is the green line and left-hand axis. Annual rainfall, mm, is the blue line and right axis. The red points are labelled for the annual percentage of the project area that was severely burned.

### 3.2. Summarised Results for Project Models

The stepwise regression modelling approach and results are illustrated in Table 3 below. Model 5, which incorporates explanatory variables for severe fire, annual rainfall and lagged dependency on previous cover levels is the best performing model structure, based on $R^2$, significance of explanatory variables and the absence of autocorrelation or a linear trend.

**Table 3.** Model results.

| 96 projects were modelled | Model 1 | Model 2 | Green Woody Cover Versus: Model 3 | Model 4 | Model 5 |
|---|---|---|---|---|---|
| | time trend counter | severe fire percentage | severe fire percentage and previous year's green woody cover (i.e., lagged dependent variable) | severe fire percentage and previous 12 months rainfall | severe fire percentage and previous year's green woody cover and previous 12 months rainfall |
| **Explanatory Variables** | | | | | |
| number of project/models which show statistical significance (Pr. < 0.1) for the severe fire variable coefficient | | 50 | 44 | 46 | 48 |
| number of project/models which show statistical significance (Pr. < 0.1) for the one-year-lagged dependent variable coefficient | | | 54 | | 54 |
| number of project/models which show statistical significance (Pr. < 0.1) for the rainfall variable coefficient | | | | 25 | 27 |
| **Diagnostics** | | | | | |
| mean value of the coefficient for the severe fire variable (significant severe fire coefficients only) | | −0.282 | −0.325 | −0.269 | −0.294 |
| number of project/models that show autocorrelation at t−1 in the residuals | 48 | 45 | 0 | 36 | 0 |
| number of project/models where a trend is found in the residuals (Pr. < 0.1) | 25 of 96 projects show a rising trend for green woody cover | 12 | 0 | 12 | 1 |
| mean value and range of the adjusted R-squared (only models with a significant severe fire coefficient) | 0.05 | 0.33 (0.13–0.75) | 0.36 (0.0–0.74) | 0.30 (0.0–0.81) | 0.41 (0.10–0.80) |

Table 3 illustrates the stepwise regression approach, progressing from left to right, with Model 5 incorporating the key ecological variables of severe fire, annual rainfall and previous level of tree cover. Figures for the explanatory variables are the number of models that achieved statistical significance for the variable of interest. Figures for the diagnostics are the number of models or the value of the diagnostic's value. A total of 96 projects were modelled. Notable is that the severe fire coefficient shows the same sign and similar magnitude when extra variables are added. Testing of each of the 96 models with the acf function in R Studio indicated no autocorrelation in the residuals for those models where the lagged dependent variable for green woody cover was included, and linear trends are eliminated from the residuals in all but one case. Although the coefficient for the one-year-lagged dependent variable was within the selected significance threshold in only 54 of the 96 models, it nevertheless improves the model.

### 3.3. Variable Coefficients' Relationship to Biophysical or Ecological Factors

To illustrate the ecological coherence of the tested explanatory variables and their coefficients, we present below plots of changes in green woody cover levels and the coefficients derived from each of the 96 project models in Model 5, compared to the long-term average rainfall gradient. Average annual rainfall, which broadly increases in a northerly direction across the study area, is an important determinant of tree cover, plant productivity and fire effects [44,70,71]. Percentage green woody cover generally increases with higher long-term average annual rainfall (Figure 3).

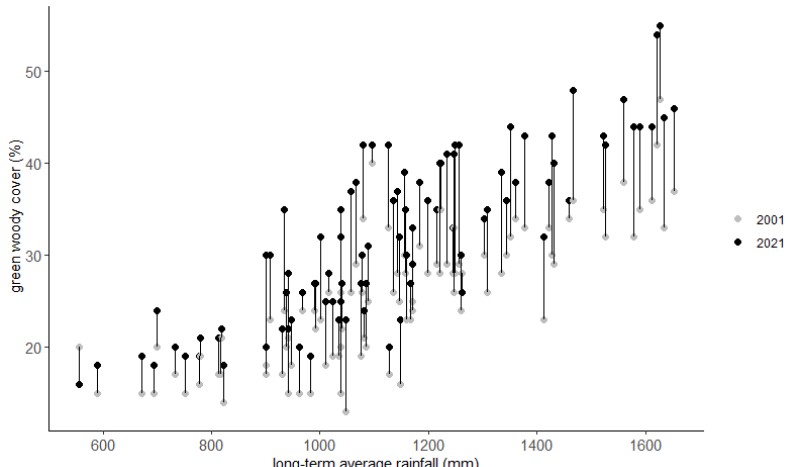

**Figure 3.** Change in green woody cover, percentage, (2001–2021) for the 96 projects, plotted against their long-term average annual rainfall.

All but two of the projects saw some increase in the fractional cover between these two years and as observed in Table 3, around one quarter of all projects show a statistically significant trend when regressing a simple time counter against annually varying green woody cover levels.

Figure 4 illustrates the values of the severe fire coefficients derived from 96 project models, against long-term average rainfall. Visual inspection may suggest that this coefficient is higher (i.e., the occurrence of severe fire causes a higher reduction in tree cover) in higher rainfall areas, which typically have higher tree cover.

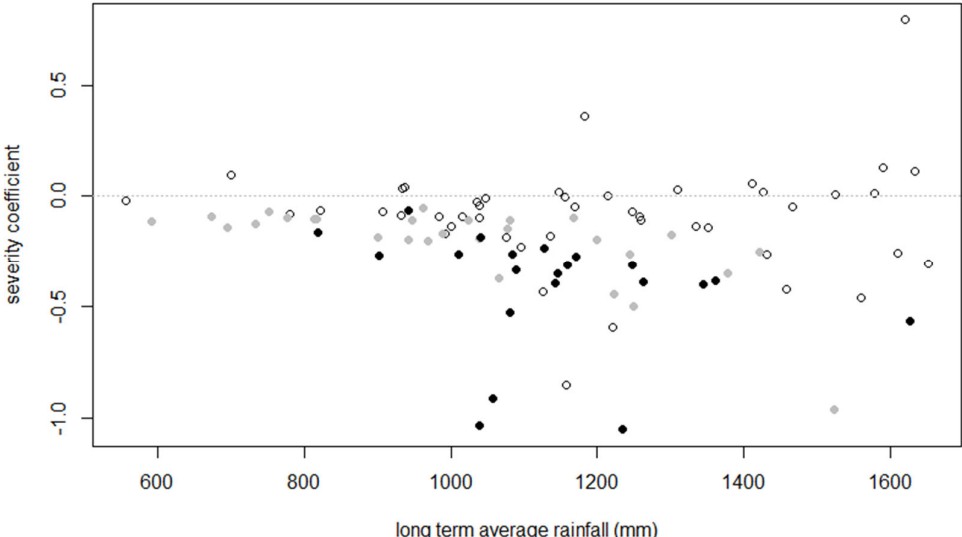

**Figure 4.** Severe fire coefficients for the 96 projects plotted against their reading for long-term annual rainfall. Filled points are those where the severe fire coefficient achieved statistical significance within their model (grey = Pr. < 0.1, black = Pr. < 0.01, unfilled= Pr. $\geq$ 0.1) Dotted line at zero is for visibility, as all coefficients are expected to be negative.

A positively signed coefficient for severe fire (above the dashed grey line at zero) is not ecologically coherent, as severe fire is not expected to lead to an increase in green woody cover. A few projects have coefficient values showing high negative outliers. These may arise when a project site experiences a large extent of severe fire in just one year, coinciding with very low levels of percentage green woody cover. This is atypical, as widespread fire is usually more common when vegetation cover is higher after strong rains. This effect was notable in 2009/2010 and may have occurred when the severe fire was early in the

dry season, and thereby had a stronger impact on the reading of green woody cover in the September–November period. By contrast, if the green woody cover rose rapidly in a year where severe fire occurred after the green woody cover reading was taken (i.e., September to November, with fire occurring in December), this could lead to a positive coefficient indicating the perverse artifact of higher vegetation cover in response to severe fire.

The annual rainfall coefficient describes the effect on the percentage of green woody cover of annually varying rainfall occurrence. This could be conceived as a measure of 'precipitation use efficiency'. As illustrated in Figure 5, in the lower average rainfall part of our study area, above average rainfall leads to increased green woody cover. However, in the high and reliable rainfall area, the coefficient is typically much lower, and in fact often negative, implying that each additional unit of rainfall leads to lower green woody cover.

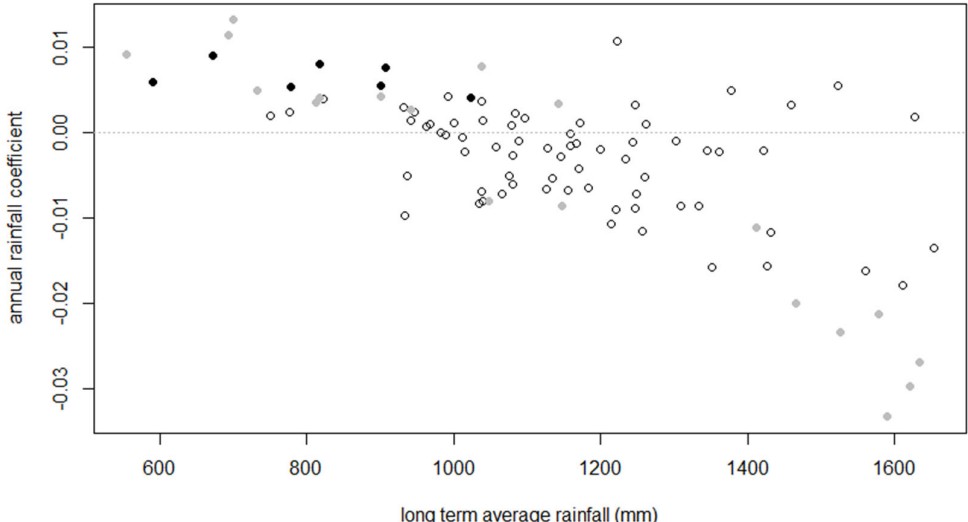

**Figure 5.** Annual rainfall coefficients for the 96 projects plotted against their reading of long-term annual rainfall. Filled points are those where the rainfall coefficient achieved statistical significance to the selected threshold (grey = Pr. < 0.1, black = Pr. < 0.01, unfilled = Pr. ≥ 0.1). Dotted line at zero is for visibility, indicating a change from negative to positive effect of rainfall.

Figure 6 below shows the estimated coefficients for the influence of the previous year's level of green woody cover, compared to parameters for severe fire occurrence and average rainfall across the study period as a whole, and compared with a measure of green woody cover level change over the study period. The filled points, typically with higher values of this coefficient, are those that achieved our selected threshold of statistical significance within their model (grey = Pr. < 0.1, black = Pr. < 0.01). The mean coefficient for the black filled points is 0.7, grey points 0.5. Adjusted $R^2$ for the lines of best fit in the three panels are 0.057, 0.067, 0.165, respectively, i.e., quite low values for the proportion of variation explained.

For some fire projects, the extent of severe fire fell substantially during the study period. The highest values for severe fire extent are typically associated with cattle properties, where late dry season burning is preferred, as this encourages new grass growth at the onset of the wet season. In this figure, there may be observed a very weak association between the size of the lagged cover coefficient and the pattern of annual rainfall, severe fire occurrence and cover change. Therefore, we are not able to draw strong conclusions about the impact of potential tree cover persistence mechanisms on either the outcome variable or tree–grass ecosystem stability.

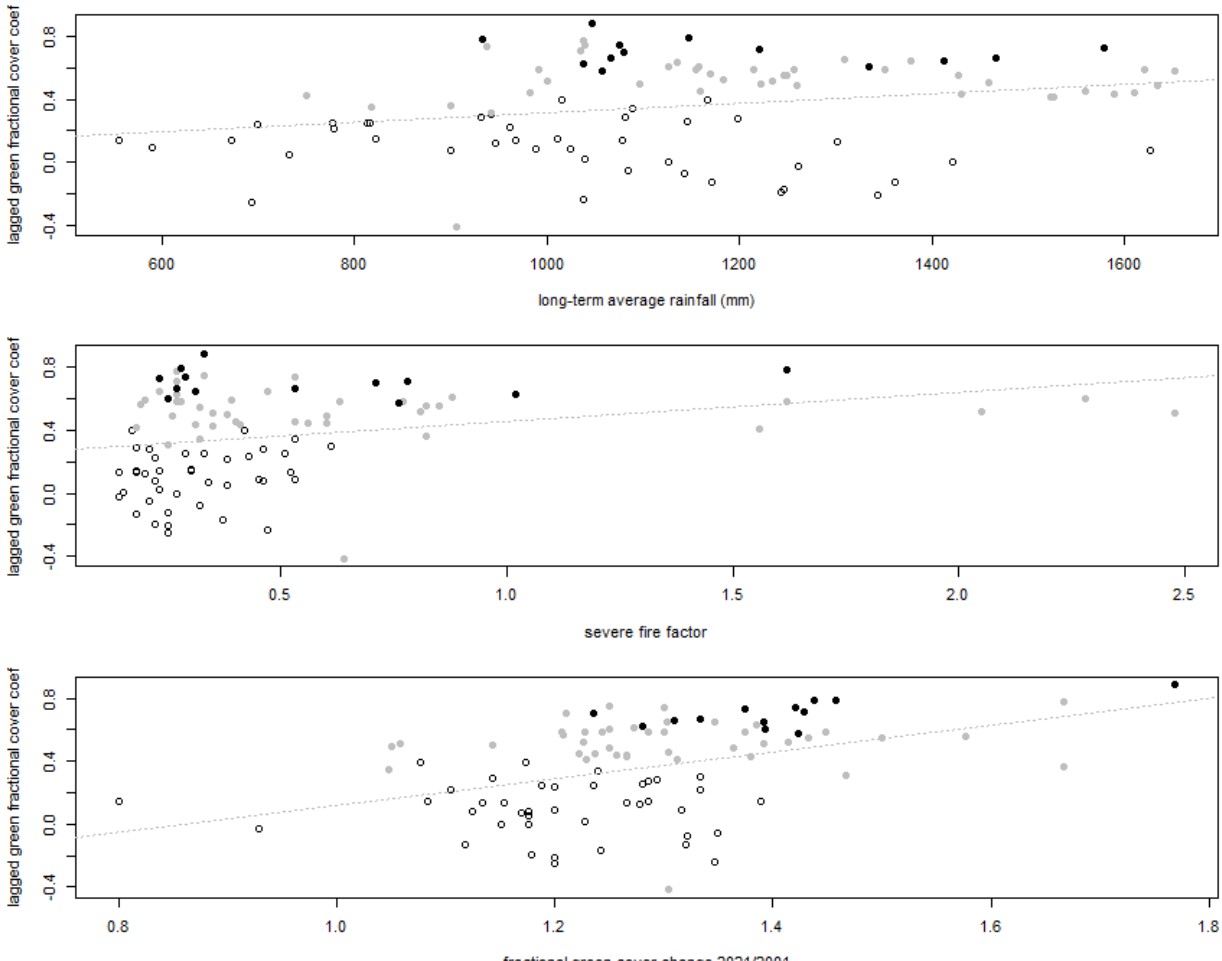

**Figure 6.** Lagged green woody cover coefficient for the 96 projects plotted against various ecological parameters. Severe fire factor approximates the average number of times a pixel within a given project would have been severely burnt over the 16-year study period. Green woody cover change is the 2021 green woody cover value divided by the 2001 value. Dotted points are a linear-regressed 'line of best fit'. Filled points are those where the lagged green coefficient achieved statistical significance to the selected threshold (grey = Pr. < 0.1, black = Pr. < 0.01, unfilled = Pr. ≥ 0.1).

## 4. Discussion

Our explanatory model of changing savanna woody vegetation cover over time demonstrates ecological effects in line with the reviewed literature. Although statistically significant coefficients for the key variables were found in only half (severe fire) or one quarter (rainfall) of the project models, when comparing all the model coefficients across the rainfall gradient, ecologically coherent explanations for these patterns emerge. Even the projects with non-significant coefficients show estimates consistent with their 'neighbours' on the long-term average rainfall gradient.

Coefficients for the negative effect of severe fire on tree cover were consistent across most models, after allowing for higher percentage green woody cover in higher rainfall zones. Rainfall coefficients vary greatly but can also be understood in the context of long-term average rainfall, with the commonly observed phenomenon of higher precipitation use efficiency in drier zones. We observed a negative coefficient for rainfall effects on tree cover levels at the wettest study sites. Whitley et al. [72] modelled seasonal and annual patterns in vegetation Gross Primary Production that correlated with solar radiation levels. They concluded that productivity at their study site, in the northern, high rainfall end of our study region, is limited by the amount of solar radiation intercepted by the canopy

rather than by water availability. This provides an explanation for how years of above average rainfall with its associated cloudiness could lead to lower green woody cover.

The 'autocorrelation' effect on green woody cover levels of previous cover levels, at a lag of one year, was eliminated by the inclusion of this lagged dependent variable in the models. In line with the broader literature on tree–grass interaction, which we briefly reviewed, the effect of this 'persistence' variable in our model is still somewhat uncertain. A coefficient of value 1 would indicate that a given percentage of tree cover, once established by disturbance or stochastic events, would persist indefinitely. Few of the coefficients approached this level, indicating that woody vegetation change is persistent in the short term but tends to revert to a stable state value over time, after controlling for the effects of the other ecological factors. Although many of these coefficients are non-significant, they can nevertheless contribute to the overall model fit and improve the reliability of other variables' coefficients.

The overall absence of linear trends after controlling for fire and rain effects suggests that factors such as changing climate effects or $CO_2$ levels, which would be expected to similarly impact vegetation across the whole of our study area, are not observable using our statistical methods. A total of 25 of the 96 properties initially showed a linear trend increase in tree cover over time. However, after the fire and rain explanatory variables are added to the model, we observed no outstanding time-based trends in the residuals, with the exception of the South-East Arnhem Land Fire Abatement Project (SEALFA) Project, which showed a residual linear trend in Model 5. This appears to be because this well-developed cool burning project has seen a reduction in the extent of both severe and non-severe fire in this project area, a phenomenon not observed in other projects.

The small number of data points, typically 16 years in our data series, means that including four explanatory terms in the model impacts strongly on the number of 'degrees of freedom' and raises the threshold for statistical significance. Work to extend the severe fire variable forward from 2018 using interpretation of the available remote sensing data would likely further improve the explanatory power of these models. A further body of work could derive an allometric that uses the measure of satellite-observable tree cover to estimate the quantity of above-ground living tree biomass [73]. This would allow the additional carbon in trees to be credited via a new measurement methodology [74] in Australia's offset scheme. Alternately, our model coefficients could be incorporated into the programming of a biomass growth model, such as FullCAM [75], which is currently used for Australia's land-based emissions inventory and for offset methodologies. Since these coefficients, especially rainfall, vary across the landscape, the approach would be to create a gridded set of modelling inputs, which is already a feature of FullCAM.

## 5. Conclusions

Using emissions offset projects from Australian tropical savannas as case studies, we have modelled satellite-observed changes in tree cover and obtained explanatory variables for the influence of severe fire occurrence, annual rainfall and the lagged effect of past percentage tree cover, in a linear regression time-series model. As new data are added to the time-series each year, the data set and model can be extended, and results will likely become more certain.

With this approach, we can separate and quantify the different drivers, including the human-controlled factors such as fire management, for any observable trend increase in savanna tree cover. Currently, any increase in living tree biomass resulting from cool burning practices is not able to be credited, as there is no approved methodology for doing so in Australia's offset scheme. Our study points to an opportunity to efficiently use remote sensing as an input to a new crediting methodology for estimating changes to carbon stocks in savannas.

As previously noted, soil, slope and fire history may all impact on the ability for a given site to store additional carbon after a change in fire regime. Our landscape scale study areas do not easily allow for the inclusion of such variables into a model, nor do we



have longer term fire histories for the sites. These factors may very well account for some of the differences in our variable coefficients. Repeating our study's methodology, using small scale sites stratified for these factors could test the significance of these factors.

**Author Contributions:** Conceptualization G.B., Data curation G.B., Methodology G.B., Writing—original draft preparation G.B., methodology G.B., Writing—Review and Editing A.E. and K.Z., Supervision K.Z. All authors have read and agreed to the published version of the manuscript.

**Funding:** This research was conducted as part of a Ph.D. thesis, at Charles Darwin University, with funding under the Research Training Program of the Commonwealth of Australia.

**Institutional Review Board Statement:** Not applicable.

**Informed Consent Statement:** Not applicable.

**Data Availability Statement:** All data used are from previously published sources described in the References section.

**Conflicts of Interest:** The authors declare no conflict of interest.

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
