# Peer review of "Fire, Rain and CO2: Potential Drivers of Tropical Savanna Vegetation Change, with Implications for Carbon Crediting"

_fire, doi:10.3390/fire6120465_

Round 1

Reviewer 1 Report

Comments and Suggestions for Authors

The manuscript entitled “Rain and CO2 potential drivers of tropical savanna vegetation change with implications for carbon crediting” conduct statics analysis on possible explanations of the increasing tree cover in savannas over Austria. The paper is well-written and I suggest to accept as it is.

Author Response

We have provided a point-by-point response to all three reviewers. Please see attachment. 

Reviewer 2 Report

Comments and Suggestions for Authors

1) Lines 43-47: What are the possible reasons for this issue? Please elaborate more details. It could be due to the variation of soil nutrients along the landform after fire. Please see this study [doi:10.3389/fenvs.2023.1213181]. Moreover, this may be due to fire frequency affect the recovery. Please see this study [doi:10.1038/nature24668].

2) Figure 1 should provide Lat/Long. It would be better to show the boundary of study in color line.

3) Line 113: “Fire frequency” How often?

4) Figure 2 is not good about the resolution and text size. Please improve.

5) Figure 3 should use difference colors between 2001 and 2021. Same comment to Figures 4-6. 

6) Line 390: Your conclusion should not provide the reference “(Clean Energy Regulator 2021b).”. Please conclude base on your findings.

7) Title should not use full stop (.).

Comments on the Quality of English Language

-

Author Response

(The authors gave the same response as above.)

Reviewer 3 Report

Comments and Suggestions for Authors

Review of Fire rain CO2

I think this could be a great article if changes are made. The authors aim is to use tree-cover data from published sources in a time series analysis to determine whether elevated CO2 levels result in increased woody vegetation cover or whether the results can be explained by precipitation and fire severity data. I have three main issues with the document. The first two are easily rectified, but the third may take additional work, thus I suggest “major” revision.

First and foremost, the authors need to do a better job of defining variables. What is “green” , “severe”, etc.?  See many notes in the .pdf.  Terms such as fire severity have very specific definitions (See, Jon Keeley (International Journal of Wildland Fire 2009) for an excellent treatment of key terms.

Second, better maps and details of study area are needed.  One of the major theoretical advances in savanna ecology in past 20 years was the division of savannas into “mesic” and semi-arid. The authors need to indicate on the map, it appears to me this is the "mesic" zone, so clearly state this, (also reference Sankaran’s original work on this). Also, the “map” is very poor. How about a map that clearly shows vegetation types at a level of specificity higher than “savanna”. Add precipitation isolines as well as these are critical.

Third, this is major, but if the authors can eliminate the use of “binaries” (e.g., LDS/EDS; severe/not severe) the study would be improved. Savannas are not binary environments (unless we are considering the tree/grass dichotomy).

For example, your precipitation data is not binary, but fire severity data is.  What effect might this have on the model? You could, for example, use fire timing (date of fire after some time 0, to create a non-binary value) given  that later fires are usually more “severe”…this is better than a binary Early/late, but still not exact since the landscape is heterogeneous (some areas will have low and other high fuel moisture on the same date).  A better metric would be fuel moisture level. Both of these variables can be derived from remote sensing imagery.  That said, if indeed you have ranches in the data then “grazing pressure” will clearly be a major actor on fuel load and thus fire intensity and severity. So perhaps references to that factor should be left out.

In summary, this paper has great potential.  I am not a statistician, so my colleagues can review the stats.  As a geographer and savanna fire scientist, I find the use of binary data very problematic.  It is unfortunate that the Australian fire science has settled on these variables. If it is not possible to use non-binary data for fire “severity” or “intensity” then the authors must provide a clear explanation for their selection of variables as well as give clear definitions. Note that in other environments (Africa, Latin America) scholars are increasingly moving away from the use of binaries to describe fire regimes (see Le Page 2010, for a very good alternative and Laris 2019 for a critique). 

My suggestion would be to use "seasonality", that is, set a time 0 and approximate fire severity as a function of time since 0.  This will capture the drying of the savanna grasses which is better correlated with intensity and severity. This way you will have precipitation and fire timing (seasonality) as two useful variables to go with "tree cover" (I would not use "greenness unless you actually calculate NDVI and I think canopy cover is what you are really testing.

Author Response

(The authors gave the same response as above.)

Round 2

Reviewer 2 Report

Comments and Suggestions for Authors

Accept in present form.

Comments on the Quality of English Language

-